There are amendments to this paper

# Modelling of free-form conformal metasurfaces

Kedi Wu [1,4], Philippe Coquet[1], Qi Jie Wang [1,2] & Patrice Genevet[3]

Artificial electromagnetic surfaces, metasurfaces, control light in the desired manner through the introduction of abrupt changes of electromagnetic fields at interfaces. Current modelling of metasurfaces successfully exploits generalised sheet transition conditions (GSTCs), a set of boundary conditions that account for electric and magnetic metasurface-induced optical responses. GSTCs are powerful theoretical tools but they are not readily applicable for arbitrarily shaped metasurfaces. Accurate and computationally efficient algorithms capable of implementing artificial boundary conditions are highly desired for designing free-form photonic devices. To address this challenge, we propose a numerical method based on conformal boundary optics with a modified finite difference time-domain (FDTD) approach which accurately calculates the electromagnetic fields across conformal metasurfaces. Illustrative examples of curved meta-optics are presented, showing results in good agreement with theoretical predictions. This method can become a powerful tool for designing and predicting optical functionalities of conformal metasurfaces for new lightweight, flexible and wearable photonic devices.

[1] CINTRA, UMI 3288, CNRS/NTU/Thales, Research Techno Plaza, 50 Nanyang Drive, Singapore 637553, Singapore. [2] Center for OptoElectronics and Biophotonics (COEB), School of Electrical and Electronic Engineering, Nanyang Technological University, Singapore 639798, Singapore. [3] Université Côte d'Azur, CNRS, CRHEA, rue Bernard Gregory, Sophia Antipolis 06560 Valbonne, France. [4] Present address: Department of Information Physics and Engineering, Nanjing University of Science and Technology, Nanjing 210094, China. Correspondence and requests for materials should be addressed to Q.J.W. (email: qjwang@ntu.edu.sg) or to P.G. (email: pg@crhea.cnrs.fr)

Metasurfaces, the two-dimensional (2D) counterparts of three-dimensional (3D) metamaterials, have attracted considerable research interest in recent years, particularly regarding their intriguing ability to control every aspect of electromagnetic waves at the subwavelength scale[1]. With respect to conventional optical devices, which progressively accumulate phase changes during light propagation along optical paths, metasurfaces introduce abrupt phase and amplitude shifts by scattering light on meta-atoms[2–13]. Taking advantage of the resonances[14–17] of metallic or dielectric nanostructures[18–23], nanoscale building blocks can control the phase and amplitude of scattered light at the subwavelength scale. By arranging the desired elements into arrays, with desired phase and amplitude profiles, one can create a plethora of metasurface devices, such as flat lenses[17], anomalous reflection/refraction deflectors[2], vortex plates[2,6,7], holograms[24], retroreflectors[25] and invisibility cloaks[26]. Due to the extreme thinness of metasurface layers compared to usual refractive phase retardant materials, electromagnetic fields can vary in a discontinuous manner across metasurfaces. The discontinuous variation of these electromagnetic fields across traditional planar metasurfaces can be modelled by considering specific boundary conditions called generalised sheet transition conditions (GSTCs)[8,27]. In this framework, metasurfaces are described as a 2D interface with abrupt surface susceptibilities, corresponding to complex reflection and transmission coefficients.

To go beyond planar metasurfaces, getting into the regime of metasurfaces with arbitrarily curved shapes, the design of free-form metasurface optical elements turns out to be remarkably complex and requires careful consideration of the substrate geometry. Recently, we have proposed a new theoretical framework called 'conformal boundary optics' to describe the electromagnetic boundary conditions at the boundaries of arbitrary geometries[28]. Given input and output field distributions, conformal boundary optics addresses the inverse engineering problems of calculating the interface response at interfaces with arbitrarily curved shapes. This model applies the concept of transformation optics at the level of the boundary conditions, transforming the electromagnetic fields expressed in the laboratory coordinate to their expression in the coordinate system conformal to the interface[29,30]. As a result, it is possible to obtain designer reflection or refraction of light from objects with unconventional shapes. This technique can be considered as a promising and revolutionary approach for designing free-form optical components and may stimulate research on new applications of metasurfaces. To conceive and evaluate the performance of the latter, new predicting numerical tools must be developed. So far, all of the published attempts to designing free-form metasurfaces have relied on the brute force approach to calculate the optical responses of a large library of individual scatterers, which are assembled side by side along non-planar devices to address the phase retardation between incident and refracted wavefronts[31]. Crude phase compensation is still not accurate at the simple level of unitary transmission through a planar device[32]. Arbitrary wavefront control requires balanced control of loss and gain or bi-anisotropic designs; quantities are generally well-captured using GSTC models. Some preliminary numerical studies, using the finite difference method and the finite element method with conventional GSTCs at planar interfaces, have been conducted[33,34], indicating the importance of research efforts on modelling in planar optics. To exploit the full potential of free-form devices, it becomes essential to develop appropriate numerical methods that are able to evaluate the performance of complex metasurfaces and connect the shapes of the interfaces with their macroscopic surface functions.

Here we propose an innovative algorithm to address free-form conformal metasurface designs. This algorithm relies on a revised finite-difference time-domain (FDTD) method, which has a great capacity for dealing with electromagnetic problems in complex geometries and inhomogeneous shapes. Commercial simulation software commonly uses standard boundary conditions with planar interfaces, such as perfect electric conductors, perfect magnetic conductors and perfectly matched layers. So far, complex boundary conditions at curved interfaces have not been implemented for conformal metasurfaces. The only exceptions are for cylinders and spheres[35,36], which are inherently conformal to cylindrical and spherical coordinate systems, respectively, and the GSTC are obtained using similar derivations. The numerical algorithm suggested in this manuscript is simple and can solve the electromagnetic fields at planar and curved interfaces. The implementation of conformal boundary optics creates new design opportunities for the next generation of conformal optical components, not only supplementing the classical theory of electrodynamics but also addressing challenging problems regarding the inversed design of new functional optical devices with desired performance. We begin our discussion by taking a set of GSTCs equations for which the reflected and transmitted fields of the incident waves are matched at planar metasurfaces. To apply these conventional GSTCs equations, additional virtual nodes have to be inserted around the interface[36,37]. Different values of surface susceptibilities tensors have been designed to mimic optical functional devices such as perfect absorbers, beam refractors and curved lenses.

## Results

**Conformal boundary optics theory of free-form metasurfaces.** We start our analysis with the traditional two-dimensional case of light transmission through a one-dimensional (1D) planar interface. Here, the metasurface is placed in the $y-z$ plane, as shown in Fig. 1a, and the electromagnetic fields are written in the Cartesian coordinate. A transverse magnetic (TM) polarised light (the magnetic field **H** is parallel to the $z$ direction) at a wavelength of $\lambda$ (frequency $\omega$) at an incident on the metasurface. The field components $H_z$, $E_x$ and $E_y$ are non-zero terms. Accordingly to the descriptions in GSTCs theory[27,28], the metasurface is treated as a zero-thickness boundary. In the general case of a planar metasurface, which reflects and refracts light at arbitrary angles, the surface electric and magnetic susceptibility tensors follow these theoretical expressions:

$$\chi_{ee} = \frac{2\sqrt{\varepsilon_0/\mu_0}(1 - t - r)}{j\omega(1 + t + r)}, \tag{1a}$$

$$\chi_{mm} = \frac{2\sqrt{\mu_0/\varepsilon_0}(1 - t + r)}{j\omega(1 + t - r)}, \tag{1b}$$

where $t$ and $r$ are the complex reflection and transmission coefficients, respectively, $\chi_{ee}$ and $\chi_{mm}$ are the electric and magnetic susceptibility tensors (first 'e/m' subscripts), respectively, and j is the imaginary unit, in response to the electric and magnetic (second 'e/m' subscripts) excitations. The theoretical descriptions transferred from the planar to the conformal metasurfaces have been proposed and discussed in[28]. In the following, we summarise the main results and equations that have been implemented in our numerical method. To simplify the analysis, we consider here only mono-anisotropic metasurface cases, for which all bi-anisotropic terms vanish $\chi_{em}=\chi_{me}=0$. The homogenised fields in the metasurface are given by the relation $\mathbf{E}=\chi_{ee}\mathbf{E_{av}}$, $\mathbf{D}=\chi_{dd}\mathbf{D_{av}}$, $\mathbf{H}=\chi_{mm}\mathbf{H_{av}}$ and $\mathbf{B}=\chi_{bb}\mathbf{B_{av}}$, (components written as $E_j = \chi_{ee}^{jk}E_{k,av}$) in which the subscript 'av' denotes the

**a**

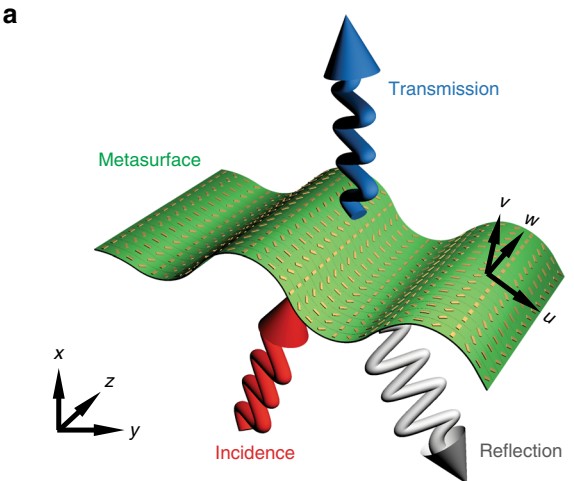

**b**

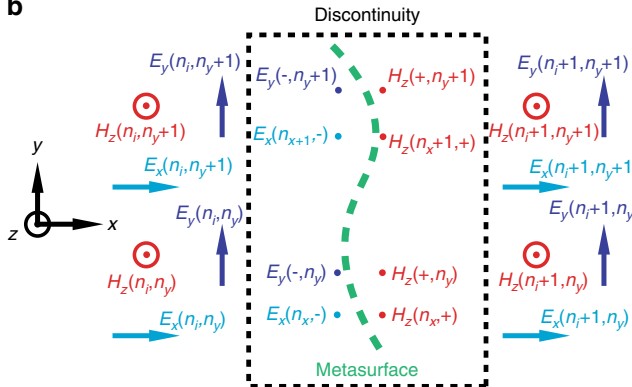

**Fig. 1** Schematic of the structure and the implementation of conformal boundary condition using FDTD algorithm. **a** Schematic of conformal metasurfaces of arbitrary geometries to modulate light propagation. **b** For non-planar geometry, conformal boundary conditions, given by Supplementary Eqs. (5) and (6) are implemented by inserting virtual nodes around to the normal Yee cell. The modification includes introducing virtual magnetic nodes $H_z^{n-1/2}(+, n_y)$ and $H_z^{n-1/2}(n_x, +)$ and virtual electric nodes $E_y^{n-1}(-, n_y)$ and $E_x^{n-1}(n_x, -)$ to calculate the field along the curved line $f'(x, y, z)$. The electromagnetic parameters of the metasurfaces are given by Eqs. (1) and (4). The field on the virtual nodes are obtained from Supplementary Eq. (2)

average of the fields taken at both sides of the metasurface. The theoretical results of the conformal boundary optics theory lead to the electromagnetic boundary conditions of the metasurface $S$ in local coordinates, written as

$$\frac{[ij]}{\sqrt{g^S}}E_j\Big|_-^+ = \frac{[ij]}{\sqrt{g^S}}\partial_j(\chi_{ee}^{wk}E_{k,av}) + \partial_t(\chi_{bb}^{ik}B_{k,av}), \quad (2a)$$

$$\frac{[ij]}{\sqrt{g^S}}H_j\Big|_-^+ = \frac{[ij]}{\sqrt{g^S}}\partial_j(\chi_{mm}^{wk}H_{k,av}) - \partial_t(\chi_{dd}^{ik}D_{k,av}), \quad (2b)$$

$$D_w\Big|_-^+ + \frac{1}{\sqrt{g^S}}\partial_i(\sqrt{g^S}\chi_{dd}^{ik}D_{k,av}) = 0, \quad (2c)$$

$$B_w\Big|_-^+ + \frac{1}{\sqrt{g^S}}\partial_i(\sqrt{g^S}\chi_{bb}^{ik}B_{k,av}) = 0, \quad (2d)$$

Using $(u, v, w)$ as the coordinate system on $S$, and $f(u, v, w)$ as a smooth function such that the surface $S$ is a level set of $f$, then we can have $g^S = \det(g_{i,j}^S) = 1 + (\frac{\partial f}{\partial u})^2 + (\frac{\partial f}{\partial v})^2$ be the Riemannian metric on the interface $S$ induced by the Euclidean norm described under the $(u, v, w)$ coordinates system, where $i=u, v$ and $k=u, v, w$ are used for Einstein's summation notation. Considering a coordinate transformation that maps the metasurface geometry from the metasurface tangent space $f(u, v, w)$ to the laboratory coordinate system $f'(x, y, z)$, we can then use conformal boundary optics to calculate the surface electric and magnetic susceptibilities. Equation (2) can be re-written in the transformed coordinate system as

$$\frac{[ij]}{\sqrt{g^S}}\Lambda_j^{k'}E_{k'}\Big|_-^+ = \frac{[ij]}{\sqrt{g^S}}\partial_j(\chi_{ee}^{wk}\Lambda_k^{k'}E_{k',av}) + \partial_t(\chi_{bb}^{ik}\Lambda_k^{k'}B_{k',av}), \quad (3a)$$

$$\frac{[ij]}{\sqrt{g^S}}\Lambda_j^{k'}H_{k'}\Big|_-^+ = \frac{[ij]}{\sqrt{g^S}}\partial_j(\chi_{mm}^{wk}\Lambda_k^{k'}H_{k',av}) - \partial_t(\chi_{dd}^{ik}\Lambda_k^{k'}D_{k',av}), \quad (3b)$$

$$\Lambda_w^{k'}D_{k'}\Big|_-^+ + \frac{1}{\sqrt{g^S}}\partial_i(\sqrt{g^S}\chi_{dd}^{ik}\Lambda_k^{k'}D_{k',av}) = 0, \quad (3c)$$

$$\Lambda_w^{k'}B_{k'}\Big|_-^+ + \frac{1}{\sqrt{g^S}}\partial_i(\sqrt{g^S}\chi_{bb}^{ik}\Lambda_k^{k'}B_{k',av}) = 0, \quad (3d)$$

where $k=u, v, w$, $k'=x, y, z$ and $\Lambda_k^{k'}$ are the Jacobian transformation matrix of the $k$ and $k'$ system. Therefore, the photonic response of the interface is as follows:

$$\chi_a^{ik'} = \begin{cases} \sqrt{g^S}\chi_a^{ik}\Lambda_k^{k'} & \text{for } i = u,v, k' = x,y,z, \\ \chi_a^{ik}\Lambda_k^{k'} & \text{for } i = w, k' = x,y,z, \end{cases} \quad (4)$$

where $a=ee, mm, dd, bb$.

**Algorithm for calculating light fields across free-form metasurfaces.** In the following discussion, we further extend the existing GSTC numerical scheme to implement time evolution of light interacting with a metasurface of arbitrary geometry. In the FDTD method, metasurfaces have the nonzero thickness $ds$, given by the spatial grid along the $x$, $y$ and $z$ directions. Considering that the metasurface is placed at $x=n_i ds$, the discretised field components are calculated using a normal FDTD update equation everywhere except at the discontinuity, i.e., along the entire simulation space except at the $(n_i$ th, $n_j$ th) node. Detailed information on these numerical methods for planar metasurface devices can be found in the following references[36–38] and a summary is presented for convenience in Supplementary Note 1 (The Derivations of Modified FDTD Equations). For a planar geometry[37], $H_z^{n+1/2}(n_i, n_y)$ and $E_y^{n+1}(n_i + 1, n_y)$ are updated by substituting the designed susceptibilities $\chi_a^{ik'}$, determined by replacing Eq. (1), into Supplementary Eqs. (5) and (6). The field distributions outside the metasurface are implemented by the normal FDTD Supplementary Eqs. (1)–(4). Substituting the values of $\chi_a^{ik'}$ into Eq. (4), the transformed susceptibilities of the curved metasurface are obtained. To account for the curvature of the interface, we make use of not only $H_z^{n+1/2}(n_i, n_y)$ and $E_y^{n+1}(n_i + 1, n_y)$ but also of introducing the additional nodes $H_z^{n+1/2}(n_x, n_j)$ and $E_x^{n+1}(n_x, n_j + 1)$. Following the same derivation procedure as in Supplementary Eqs. (5) and (6), $H_z^{n+1/2}(n_x, n_j)$ and $E_x^{n+1}(n_x, n_j + 1)$ are calculated with the

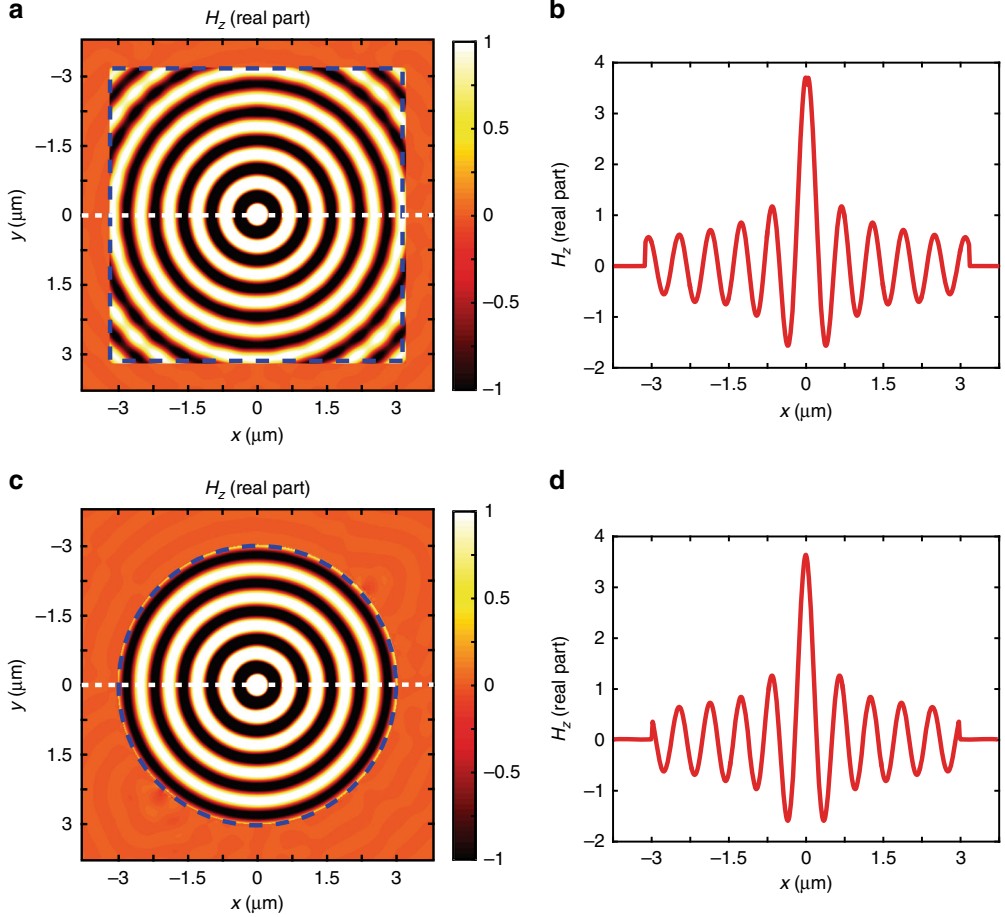

**Fig. 2** Arbitrary shaped perfect absorbing layers. Results of numerical simulation showing the magnetic field intensity distribution (real part of $H_z$ field) of a point source located in $x = 0$, $y = 0$. The emitted light propagates towards a perfectly absorbing metasurface. **a** Magnetic field intensity distributions of planar absorbing metasurface disposed along the dashed blue rectangle. **c** light field intensity distributions of cylindrical absorbing metasurface disposed along the blue dashed circle. Both metasurface geometries—planar and conformable—present high extinction ratio and negligible reflection, indicating that metasurface boundary conditions can be used as ultrathin perfect absorbing layers. **b** and **d** show the cross profile of the field distribution along the $x$ axis at plane $y = 0$ as indicated by the white dashed line in **a** and **c**

following expressions:

$$
\begin{aligned}
H_z^{n+1/2}(n_x, n_j) &= \frac{1 - j\omega\chi_{mm}^{wz}dt/4ds}{1 + j\omega\chi_{mm}^{wz}dt/4ds} H_z^{n-1/2}(n_x, n_j) \\
&+ \frac{dt}{\mu_0 ds(1 + j\omega\chi_{mm}^{wz}dt/4ds)}(E_x^n(n_x, n_j + 1) - E_x^n(n_x, n_j) \\
&+ E_y^n(n_x, n_j) - E_y^n(n_x + 1, n_j)) \\
&- \frac{j\omega\chi_{mm}^{wz}dt/4ds}{1 + j\omega\chi_{mm}^{wz}dt/4ds}(H_z^{n+1/2}(n_x, n_j + 1) + H_z^{n-1/2}(n_x, n_j + 1)),
\end{aligned}
\tag{5}
$$

and

$$
\begin{aligned}
E_x^{n+1}(n_x, n_j + 1) &= \frac{1 - j\omega\chi_{ee}^{ux}dt/4ds}{1 + j\omega\chi_{ee}^{ux}dt/4ds} E_x^n(n_x, n_j + 1) \\
&+ \frac{dt}{\varepsilon_0 ds(1 + j\omega\chi_{ee}^{ux}dt/4ds)}(H_z^{n+1/2}(n_x, n_j + 1) - H_z^{n+1/2}(n_x, n_j)) \\
&- \frac{j\omega\chi_{ee}^{ux}dt/4ds}{1 + j\omega\chi_{ee}^{ux}dt/4ds}(E_x^n(n_x, n_j) + E_x^{n+1}(n_x, n_j)).
\end{aligned}
\tag{6}
$$

Note that Eqs. (5) and (6) represent dispersion-less interfaces. The effects of dispersion have been treated for planar interface in[38]. The overall field distribution is then calculated along

arbitrary geometry by updating fields considering Supplementary Eqs. (5) and (6) and Eqs. (5) and (6).

**Arbitrarily shaped perfectly absorbing metasurfaces**. To validate our FDTD-based numerical algorithm based on the conformal boundary optics theory, we propose solving some examples in a two-dimensional space. The first example is a perfectly absorbing planar metasurface placed at $z = 0$, suspended in the air, as shown in Fig. 2a. The illumination is a TM polarised light which propagates to the interfaces at a wavelength of $\lambda = 600$ nm. The spatial grid step is $ds = 15$ mm, and the time step is $dt = 2.5\text{e}{-}17\text{s}$. An ideal absorbing surface is defined by $r = t = 0$, which, according to Eq. (1) on the layer susceptibilities, leads to $\chi_{ee} = 2\sqrt{\varepsilon_0/\mu_0}/j\omega$ and $\chi_{mm} = 2\sqrt{\mu_0/\varepsilon_0}/j\omega$. For the purpose of launching a broad distribution of transverse momentum at the interface, we place a point source in the centre of the simulated area and place the absorbing metasurfaces to form a 6.3 μm × 6.3 μm rectangular frame surrounding the point source (blue dashed line in Fig. 2a). The uniaxial perfect matched layer (UPML) is used as the conventional absorbing boundary conditions on the outermost regions of the simulated area (7.5 μm × 7.5 μm for 600 nm wavelength), i.e., right after passing through the metasurface in the dashed blue line. The UPML thickness of 0.27 μm (~$\lambda/2$) surrounding the simulation area has been chosen to cancel

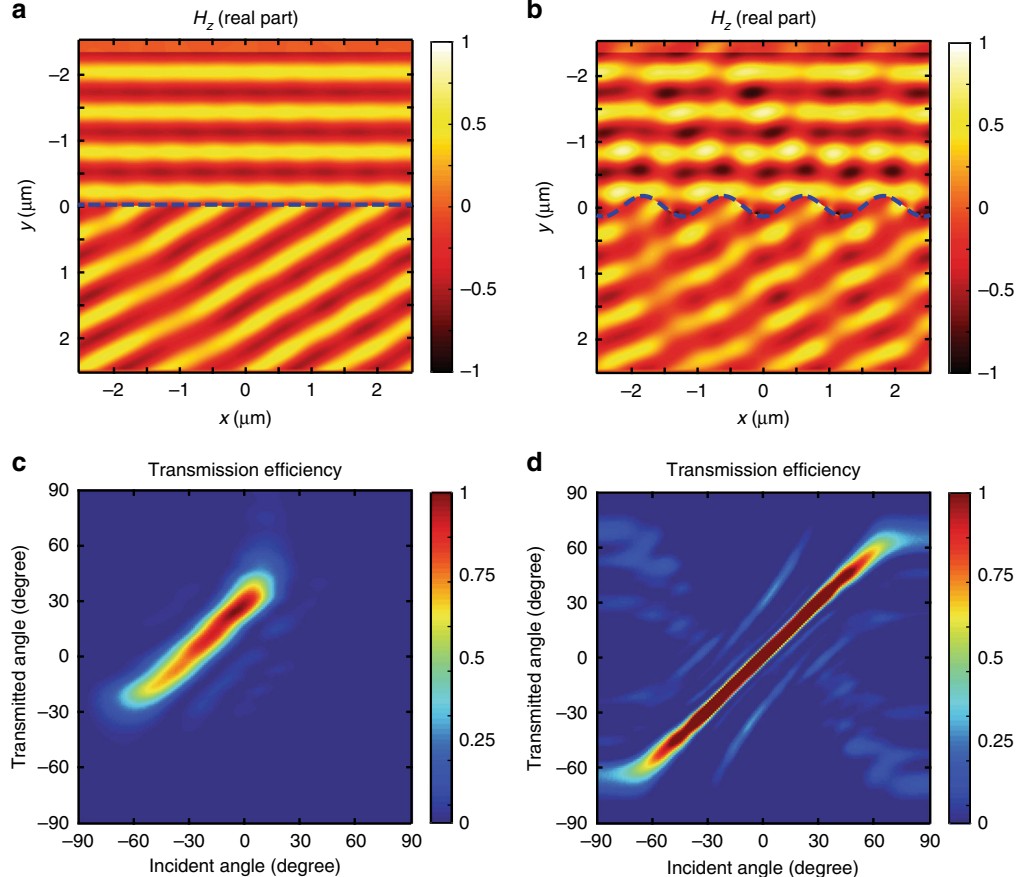

**Fig. 3** Non-planar metasurface beam deflectors. Results of numerical simulation showing the magnetic field intensity distribution of the beam deflector design, according to the generalised Snell laws, to refract light to an angle of $\pi/6$ rad. In the calculations, the incident light is a TM polarised plane wave propagating along the $y$-direction. **a** and **b** Grey intensity distributions of planar and sinusoidal beam deflector. Blue dashed lines indicate the position and geometry of the interfaces. **c** Transmission efficiency of the sinusoidal metasurface in **b** as a function of the incident angle. **d** Transmission efficiency of a conventional sinusoidal grating for comparison to **c**

backward reflection, mimicking free-space outgoing propagation. Figure 2a shows the magnetic field distribution after light propagation through the absorbing metasurface as defined by Supplementary Eqs. (5) and (6). We observe that the wavefront inside the region enclosed by the metasurface is exactly equivalent to free-space propagating light in the air, meaning that reflection is cancelled along the metasurface boundaries. Beyond the metasurface, the amplitude of the transmitted fields is on an order of magnitude smaller (1e−5) than the incident light. These observations ($r=0$, $t=0$) indicate that the incident light has been absorbed by our simplified version of perfectly absorbing metasurfaces. To illustrate quantitatively the amplitude of the fields inside and beyond the metasurface boundaries, we present the cross line profile of the field distribution following the white dashed line along the $x$ axis at $y=0$, as seen in Fig. 2a. The result in Fig. 2b already indicates that the waves are perfectly absorbed by a single layer metasurface, which may have interesting numerical application in replacing the PML in conventional FDTD simulations, having the advantage of significantly reducing computational time. For comparison, conformal boundary optics calculations have been performed to mimic, in example 2, a perfectly absorbing circular metasurface. The chosen circular metasurface is described by expression $f(R, \theta, n)=R-10\lambda$. We impose the same perfectly absorbing condition, leading to the following susceptibilities for the cylindrical layer $\chi_{ee}^{rr} = \chi_{ee}^{\theta\theta} = 2\sqrt{\varepsilon_0/\mu_0}/j\omega$ and $\chi_{mm}^{zz} = 2\sqrt{\mu_0/\varepsilon_0}/j\omega$. From Eq. (4), the susceptibilities in the ($x$, $y$, $z$) coordinate system is $\chi_{ee}^{rx} = \chi_{ee}^{\theta y} =$

$2\sqrt{\varepsilon_0/\mu_0}/j\omega$ and $\chi_{mm}^{zz} = 2\sqrt{\mu_0/\varepsilon_0}/j\omega$. The field distribution of light passing through an interface of radius $R=3$ μm is shown in Fig. 2c. As with the planar case, the reflected and transmitted light are zero after interacting with our designer curved absorbing metasurface. This observation is also supported by quantitative measurements of the cross line profile of the field distribution along the $x$ axis at plane $y=0$ (Fig. 2d). We observe that $H_z=0$ when $R>10\lambda$, meaning that we obtain perfect absorption for the metasurface interface.

**Non-planar metasurface beam deflectors**. The next example consists of a planar gradient phase at the interface to anomalously refract light in a predefined direction[1]. Early studies have extensively used these gradient phase metasurfaces to validate metasurface designs. In our example, a planar metasurface (parallel to the $x$-direction) is designed to refract a normally incident light (propagating along the $y$-direction) to a plane wave propagating on the other side of the metasurface at an angle $\theta_t=\pi/6$ to its original direction. The interface response is defined by considering the reflection coefficient $r=0$ and the complex transmission coefficient $t=\exp[j2\pi\sin(\theta_t)x/\lambda]$. This way, the susceptibilities of such metasurfaces are given by the following: $\chi_{ee}^{yy} =$

$\frac{2\sqrt{\varepsilon_0/\mu_0}\{1-\exp[j2\pi\sin(\theta_t)x/\lambda]\}}{j\omega\{1+\exp[j2\pi\sin(\theta_t)x/\lambda]\}}$ and $\chi_{mm}^{zz} = \frac{2\sqrt{\mu_0/\varepsilon_0}\{1-\exp[j2\pi\sin(\theta_t)x/\lambda]\}}{j\omega\{1+\exp[j2\pi\sin(\theta_t)x/\lambda]\}}$. Substituting these susceptibilities into Supplementary Eqs. (5) and (6) and carrying out FDTD simulations, we obtain the magnetic

field distributions shown in Fig. 3a. We observe that the incident light is refracted to the desired angle of $\pi/6$ and that the reflection is almost negligible. To validate our numerical scheme, we calculated and implemented the surface susceptibilities to impose anomalous refraction of light at curved interfaces, i.e., a curved beam refractor, also designed to refract light at $\theta_t = \pi/6$. The values of the susceptibilities used for this calculation are presented in Supplementary Fig. 2. In the presence of metasurfaces, the relation between the incoming and the outgoing waves can be imposed at the interface by considering the condition that the difference between the propagation phase shift of two light rays of the incident wavefront impinging on the surface at points (separated by a distance $(\delta u, \delta v)$), and the propagation phase shift after interface $\delta\varphi_t$ is exactly compensated by the phase shift introduced at the interface at those points. However, this is a crude approximation, and to properly account for amplitude, and for phase and polarisation discontinuities, the local orientation of the interface has to be considered and the boundary conditions have to be satisfied for all of the orientations. Here, conformal boundary optics becomes a handy solution to obtain these surface susceptibility tensors. Considering a sinusoidal surface with a surface function $f(u, v, w) = 0.75\lambda\cos(\pi u/\lambda) - u$, giving the interface position and supposing that the refraction in the curved coordinate system is $\theta_t' = \pi/6 - \tan^{-1}\left(\frac{\partial f(u,v)}{\partial(u,v)}\right)$, we obtain from Eq. (4) the following susceptibilities: $\chi_{ee}^{ux} = \chi_{ee}^{vy} = $

$$\frac{2\sqrt{\varepsilon_0/\mu_0}\{1 - \exp\{j\{2\pi\sin(\theta_t)x/\lambda + 1.5\pi\cos(\pi x/\lambda)[\cos(\theta_t) - 1]\}\}\}}{j\omega\{1 + \exp\{j\{2\pi\sin(\theta_t)x/\lambda + 1.5\pi\cos(\pi x/\lambda)[\cos(\theta_t) - 1]\}\}\}}$$ and

$\chi_{mm}^{wz} = \frac{2\sqrt{\mu_0/\varepsilon_0}\{1 - \exp\{j\{2\pi\sin(\theta_t)x/\lambda + 1.5\pi\cos(\pi x/\lambda)[\cos(\theta_t) - 1]\}\}\}}{j\omega\{1 + \exp\{j\{2\pi\sin(\theta_t)x/\lambda + 1.5\pi\cos(\pi x/\lambda)[\cos(\theta_t) - 1]\}\}\}}$. To avoid wave trapping and subsequent multiple reflections on the metasurface, the oscillation period of the considered surface function is larger than the wavelength of incident light. Further investigations are required to study subwavelength oscillation regimes, a topic that is beyond the scope of this paper. The magnetic field distribution of a normally incident light after interaction with our sinusoidal beam refractor is given in Fig. 3b. After analysing the results shown in Fig. 3a, b, we conclude that the same refraction effects can be achieved for interfaces with curved geometries. We also calculated the refraction efficiency of the sinusoidal beam refractor by sweeping the angle of incidence $\theta_i$ from $-\pi/2$ to $\pi/2$. Collecting the electromagnetic fields immediately after the curved interface, we apply the near-field to far-field transformation[32] and calculate the far-field refraction efficiency, as presented Fig. 3c.

The colour map indicates that the maximum transmission efficiency occurs for a normally-incident angle $\theta_i = 0$ and a refracted angle of $\pi/6$, which agrees with the theoretical prediction and the design. As has been pointed out in[31,39,40], the use of mono-anisotropic expressions for the surface susceptibilities leads to the appearance of spurious background signals that result from higher diffraction orders. According to the literature, background-free and exact results are obtained by considering a bi-anisotropic metasurface, which is also applicable to our method. In the absence of such a metasurface, a sinusoidal surface diffracts light into plane waves propagating at angles $\theta_t$ given by the grating formula $m\lambda = \Gamma(n_i\sin\theta_i - n_t\sin\theta_i)$, where $\Gamma$ is the oscillation period. For the purpose of comparison with such conventional grating, we plotted the diffraction efficiency of a sinusoidal grating made of silica with a refractive index $n = 1.5$ and having the same interface geometry, $f(u, v, w) = 0.75\lambda\cos(\pi u/\lambda) - v$, as the above curved metasurface. Figure 3d shows the diffraction efficiency of all three of the available diffraction orders, one of which can partly bend light to angle $\pi/6$ but not for incident light at $\theta = 0$.

**From flat to curved meta-lenses.** The previous examples showed that plane waves can be directed along a specific direction. However, as we demonstrate, a similar approach can be used to design more complicated optical elements. This is generally achieved by introducing a complex phase profile on the metasurface. As examples, two types of lenses that focus light with focal length $l = 2.5\,\mu m$ are demonstrated. For a planar lens with complete transmission, the reflection coefficient $r = 0$ and the complex transmission coefficient $t = \exp(j\pi x^2/\lambda l)$ result in the susceptibility tensors as $\chi_{ee}^{yy} = \frac{2\sqrt{\varepsilon_0/\mu_0}[1 - \exp(j\pi x^2/\lambda l)]}{j\omega[1 + \exp(j\pi x^2/\lambda l)]}$ and $\chi_{mm}^{zz} = \frac{2\sqrt{\mu_0/\varepsilon_0}[1 - \exp(j\pi x^2/\lambda l)]}{j\omega[1 + \exp(j\pi x^2/\lambda l)]}$. Performing our simulations, the light field distributions behind the planar lens were obtained and are presented in Fig. 4a. As expected from a flat lens, it is shown that the light is focused at the point $(0, 2.5\,\mu m)$. Considering a sinusoidal curved lens, and applying the above discussed design procedure by using Eq. (4), we derive the susceptibilities of the curved lens as $\chi_{ee}^{ux} = \chi_{ee}^{vy} = \frac{2\sqrt{\varepsilon_0/\mu_0}\{1 - \exp\{j\{\pi x^2/\lambda l + 1.5\pi\cos(\pi x/\lambda)[\cos(\theta_t) - 1]\}\}\}}{j\omega\{1 + \exp\{j\{\pi x^2/\lambda l + 1.5\pi\cos(\pi x/\lambda)[\cos(\theta_t) - 1]\}\}\}}$ and $\chi_{mm}^{wz} = \frac{2\sqrt{\mu_0/\varepsilon_0}\{1 - \exp\{j\{\pi x^2/\lambda l + 1.5\pi\cos(\pi x/\lambda)[\cos(\theta_t) - 1]\}\}\}}{j\omega\{1 + \exp\{j\{\pi x^2/\lambda l + 1.5\pi\cos(\pi x/\lambda)[\cos(\theta_t) - 1]\}\}\}}$. The values of the susceptibilities used for this calculation are presented in

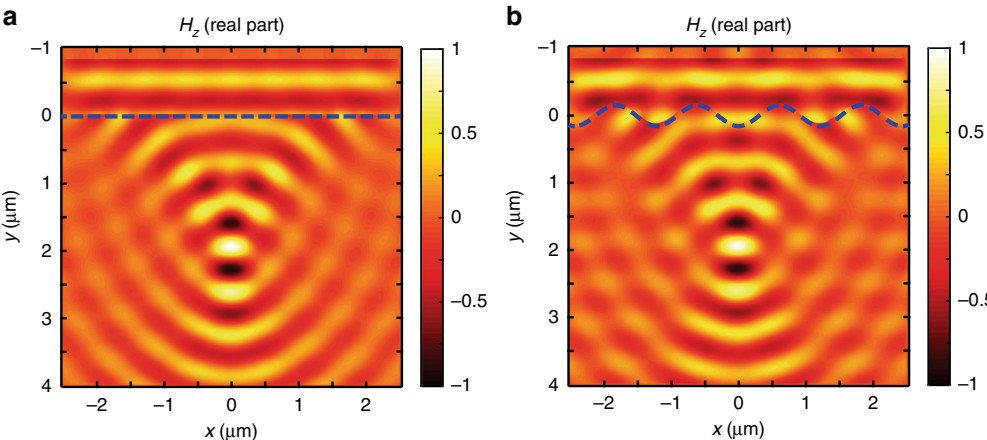

**Fig. 4** From flat to curved meta-lenses. Magnetic field intensity distribution of light transmitted through a planar **a** and a free-form **b** lens with focal of 2.5 μm. In the calculations, the incident light is a TM polarised plane wave propagating along the $y$-direction. Blue dashed lines indicate the interfaces. These results show that conformal boundary optics is successfully implemented in a modified FDTD numerical scheme. It offers rapid numerical testing of free-form optical devices in view of their optimisation and benchmarking with planar metasurface devices

Supplementary Fig. 3. The magnetic field distribution in Fig. 4b shows the focusing effect of such a sinusoidal metasurface, which agrees well with the simulation in Fig. 4a, and which suggests that our numerical scheme is able to handle complex interface geometries and arbitrarily abrupt field discontinuities for designing new types of optical devices.

## Discussion

It is worth noting that it is indeed possible to calculate curved problems considering locally flat interfaces. However, the local condition for matching incident and outgoing waves may have to be adjusted locally, i.e., adjusting the incident angle with respect to the surface orientation and recalculating the susceptibilities point-by-point along the interface. In essence, this is exactly the purpose of using the conformal boundary condition. With this method, one can calculate the photonic response of the interface in the surface coordinate, which is intrinsically flat in the interface coordinate system. Then one can transform both incoming and outgoing electromagnetic fields and susceptibilities from the local to the global coordinate system. This method allows us to simplify the problem at the conceptual level and to significantly reduce the problem of coding locally the boundary conditions.

In summary, we propose and demonstrate a new FDTD modelling method to simulate and design free-form metasurfaces of arbitrary shapes and functionalities. The discontinuities in electromagnetic fields across metasurfaces have been properly considered and implemented using a new numerical formulation. Moving from the planar to the conformable optical design, we implemented ultrathin perfectly absorbing layers, artificial gratings and curved metalenses by using conformal metasurfaces with a performance identical to their planar counterparts. In addition, our proposed method can be extended to three-dimensional cases and bi-anisotropic metasurfaces. Our method can also be used as a simple and efficient platform to design and demonstrate new lightweight, small-scale and/or wearable optical devices[41] with strong potential applications in the fields of military camouflage, non-invasive imaging in the life sciences and optical encryption in information processes, among others.

**Data availability**. The data that support the plots and other findings in this study are available from the corresponding author upon reasonable request.

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

## Acknowledgements

P.G. gratefully acknowledges financial support from the European Research Council (ERC) under the European Union's Horizon 2020 research and innovation programme (grant agreement FLATLIGHT No 639109). Q.J.Wang acknowledges the Singapore Ministry of Education Tier 2 Program (MOE 2016T2-1-128) and the Singapore National Research Foundation Competitive Research Program (NRF-CRP18-2017-02). K. Wu is also supported by the National Natural Science Foundation of China (NSFC) (Grant No. 11504243), the Natural Science Foundation of Guangdong Province, China (Grant Nos. 2016A030313042 and 2015A030310400) and the Fundamental Research Funds for the Central Universities (No. 30918011337).

## Author contributions

P.G. conceived the idea of this study. P.G. and Q.J.W. directed the research. K.W. performed the theoretical analysis and designed the core algorithms for simulation. K.W., P.C., Q.J.Wang and P.G. prepared the manuscript. Q.J.W. and P.G. supervised the project. All of the authors discussed the results and commented on the manuscript.

## Additional information

**Competing interests:** The authors declare no competing interests.

