## [Peer Review File · Nature Communications]

Reviewers' comments:

Reviewer #1 (Remarks to the Author):

GENERAL COMMENTS

The paper presents a smart and useful technique to analyze curved metasurfaces with FDTD. It is definitely worth publishing, but has a two main issues that must be fixed before that.

1. The first issue is the lack of recognition that all the part on the FDTD-GSTC has already been published. The authors seem to have missed the following papers:

arXiv:1701.08760v2 [physics.class-ph] 11 May 2017 (available for preview with IEEE T-AP)

arXiv:1710.00044v1 [physics.app-ph] 14 Sep 2017

arXiv:1710.11264v1 [physics.optics] 30 Oct 2017

Everything until line 161, with Figs. 1(b) and 1(c), has been present in the above papers, and even already in the first of the three (May 2017), including the idea of the virtual nodes to incorporate GSTCs without change the FDTD core algorithm, except for some very minor variations. This must naturally be properly acknowledged. The only novelty with regard to FDTD-GSTC is the concept of introducing virtual nodes along two directions to accommodate the curved metasurface, as done at p. 9, related to Figs. 1(d).

2. The second issue is that the paper lacks clarity. Based on comment 1, the essential contribution of the paper is the combination of GSTC-FDTD with coordinate transformation (called here conformal optics or transformation optics; incidentally, since the paper is about a method "coordinate transformation" would be a better terminology, as this has already been used and reported in the computational electromagnetics literature). This is partly based on the excellent work on the authors on "conformal boundary optics" [28]. However, the manuscript gives only one equation about it [Eq. (8)] and not enough related information for self-consistency. We suggest to reduce some parts related to the GSTC-FDTD, replacing them with appropriate references (comment 1), and instead use the available space to better describe the GSTC-FDTD + Coordinate Transformation procedure, which is the real core and contribution of the paper.

I have performed this review with the help of a student working in this area, and we both have difficulties in understand the gist of the method and a number of specific elements. The key idea of the paper is to use the conventional rectangular Yee grid (in contrast to what is erroneously stated somewhere, see below) everywhere, without stair-case approximation of the curved surface, and that the curvature of the metasurface is completely absorbed in the coordinate-transformed susceptibility functions. The "conformal" GSTCs are applied locally with curvature information included exclusively in them and not anywhere else. This fact and precise description of corresponding steps are not clear enough in the paper. Moreover, several points lack clarity and accuracy, some of which are mentioned below.

DETAIL COMMENTS

1. There are many grammatical errors. To be carefully revised and corrected.

2. Lines 77-78 read: "...complex boundary conditions with curved interfaces have never been implemented". This is incorrect. GSTCs have been applied to cylindrical and spherical metasurfaces in the following two respective papers:

- M. Safari, A. Abdolali, H. Kazemi, M. Albooyeh, M. Veysi, and

F. Capolino, "Cylindrical metasurfaces for exotic electromagnetic wave manipulations," in 2017 IEEE International Symposium on Antennas and Propagation USNC/URSI National Radio Science Meeting, July 2017, pp. 1499–1500.

- arXiv:1710.00040v1 [physics.app-ph] 7 Sep 2017

This must be correct and these reference must be added. Their existence does not reduce in any

ways the merits of the proposed method, since those 2 papers work only on canonical (circular cylinder and spherical) curved metasurface, whereas the proposed method can handle any curvature.

3. The aforementioned "any curvature" is not completely right. For instance in example 2 (sinusoidal metasurface), the ratio between the modulation height and structure pitch must be small enough to avoid wave trapping and subsequent multiple reflections. A note on such restrictions should be included!

4. Detail: note, in relation to the synthesis (7), that diffraction-less refraction cannot be achieved in a monoanisotropic metasurface, as shown in arXiv:1705.09286v2 [physics.class-ph] 12 Jun 2017: a bianisotropic metasurface is required for that. This is a detail that does not affect the essence of the paper (on the contrary, the result show the spurious diffraction order, weak here due to the small angle), but this may be noted by the authors.

5. Eq. (8): $a = ab$ is impossible unless $a = 0$. Please, use different notation for the left-hand side χ and the χ involved in the right-hand side.

6. Fig. 1: draw the local coordinate system for convenience. Also correct coordinate inconsistencies between (a) and (b).

7. Lines 223-262: correct the order of the superscripts.

8. Line 400 and other places in the paper refers to "modified Yee cell". This is wrong, or at least misleading (see general comment 2 above). The Yee grid/cell is unchanged; only the "welding" update equations between REGULAR Yee cells are changed.

9. Examples should be described with a bit more clarity.

Reviewer #2 (Remarks to the Author):

In this paper, the authors proposed a modified finite difference time-domain (FDTD) approach to calculate the electromagnetic fields across conformal metasurfaces of arbitrary shapes. The reviewer believes it is a good effort to investigate new computational methods to extend conventionally metamaterial simulation from Euclidean space to Riemann geometry. The proposed method may be utilized for the simulation and design of conformal metasurfaces. The reviewer recommends accepting the paper for publication in Nature Communications after the following problem has been fixed.

(1) The authors should better elaborate the advantages of the modified FDTD methods versus the conventional FDTD method. Since its introduction by Yee, FDTD has already been widely used to obtain numerical solutions of Maxwell's equations for a broad range of problems. In the current content, it is not very clear what new improvements have been made in the modified FDTD.

(2) The authors claim that the proposed method can "simplify and accurately calculate the electromagnetic fields across conformal metasurfaces of arbitrary shapes". It is not clear to the reviewer how the proposed method can model complex structures using regular lattice grids and how to capture the discontinuity of material properties across the interface of different materials in the metasurfaces. This is an fundamental issue in the design of multimaterial metasurfaces. It will be very helpful if the authors can better elaborate it with an example.

(3) Realistic FDTD simulations involve fine discretization of the spatial domain as well as the temporal domain. It is not uncommon to use a huge number of spatial grids and spatiotemporal grids. As a result, FDTD simulations require significant computational resources both in terms of memory and execution. It would be helpful if the authors can better elaborate the efficiency of the proposed method and whether is scalable to large-scale problems.

Reviewer #3 (Remarks to the Author):

The authors propose a new numerical method based on modified finite difference time domain approach that simplifies and accurately calculate the electromagnetic fields across conformal metasurfaces or arbitrary shapes. To validate this method, the authors propose 3 different examples such as absorbing metasurfaces, beam refractor, and curved lens, showing consistent results in agreement with fully theoretical predictions.

Although the paper is interesting, in my opinion, it does not introduce any novel phenomenon and does not provide any performance or technological advancement with respect to previous works based on standard FDTD simulations by using dense mesh, I believe it does not meet the impact and novelty requirements to be published in Nature communications journal.

It is well known that the conventional FDTD need to be refined when dealing with curved surfaces for accurate results. In fact, modeling dispersive materials with curved surfaces still remains a challenging topic, not only because the algorithm dealing with it is complex, but also because it can become numerically unstable. One approach to addressing this problem is based on the use of effective permittivities (EPs). This method is well known in the literature (see reference below). M. G. Silveirinha, A. Alu, and N. Engheta, "Parallel-plate metamaterials for cloaking structures," *Phys. Rev. E*, 75, 036603 (2007).

G. W. Milton and N. P. Nicorovici, "on the cloaking effects associated with anomalous localized resonance," *Proc. R. Soc. A*, 462, 3027--3059 (2006).

D. Schurig, J. B. Pendry, and D. R. Smith, "Calculation of material properties and ray tracing in transformation media," *Opt. Express*, 14, 9794--9804 (2006).

On the other hand, this kind of structure is well understood, and a numerical and experimental paper has been published recently in Nature Communications without using this method.

"Decoupling optical function and geometrical form using conformal flexible dielectric metasurfaces" *Nature Communications* 7, 11618 (2016)

In my opinion, the term "a new and powerful design tool urgently needs it" is exaggerated.

In order to improve and give more impact to the paper for other journals like the journal of computational physics, physical review E.

I suggest the authors address the following questions.

1. In this paper, equation 7 presents the relation between surface electric and magnetic susceptibility tensor and reflection and transmission coefficients. Although it is a common equation in GSTCs theory, I still suggest the authors provide a reference. Besides, this equation is only for normal incidence. I also suggest the author mention it in the paper or expand the method to oblique incidence

2. In this paper, the authors demonstrated three examples, absorber, deflector, and concentrator with flat case and curved boundary condition. Also, the curved boundary condition can be also solved by locally flat boundary condition with small mesh. In order to see the difference between the old (flat) and the new (curved) method, I suggest the author compare the results of them.

3. In figure 3(b), the variation of the wave front is much larger than the flat case in figure 3(a). I suggest the authors decrease the mesh size. It is also the same case in figure 4(b).

***Response to the comments of Reviewer 1.**

REVIEWER GENERAL COMMENTS

The paper presents a smart and useful technique to analyze curved metasurfaces with FDTD. It is definitely worth publishing, but has two main issues that must be fixed before that.

1) The first issue is the lack of recognition that all the part on the FDTD-GSTC has already been published. The authors seem to have missed the following papers:

arXiv:1701.08760v2 [physics.class-ph] 11 May 2017 (available for preview with IEEE T-AP)

arXiv:1710.00044v1 [physics.app-ph] 14 Sep 2017

arXiv:1710.11264v1 [physics.optics] 30 Oct 2017

Everything until line 161, with Figs. 1(b) and 1(c), has been present in the above papers, and even already in the first of the three (May 2017), including the idea of the virtual nodes to incorporate GSTCs without change the FDTD core algorithm, except for some very minor variations. This must naturally be properly acknowledged. The only novelty with regard to FDTD-GSTC is the concept of introducing virtual nodes along two directions to accommodate the curved metasurface, as done at p. 9, related to Figs. 1(d).

First, we would like to thank the reviewer for dedicating the time and the efforts to read and comment on our article. Your comments are very valuable and we believe that they helped us improving our manuscript.

Concerning the comments. We thank the reviewer for recommending us the important papers about the GSTC-FDTD algorithm in the case of planar optical devices. We were not aware of two of the most recent arxiv papers. We have addressed this point and cited these papers in the revised manuscript. The descriptions about the conventional planar GSTC-FDTD method in the original manuscript have been removed from the main article and transferred to the supplemental materials. We followed your advice to replace the previous discussion with a self-consistent introduction on conformal boundary optics that includes the theoretical calculations implemented in the algorithm. Thank you for this comment, it certainly strengthen the content of our manuscript and make it more focused.

2) The second issue is that the paper lacks clarity. Based on comment 1, the essential contribution of the paper is the combination of GSTC-FDTD with coordinate transformation (called here conformal optics or transformation optics; incidentally, since the paper is about a method “coordinate transformation” would be a better terminology, as this has already been used and reported in the computational electromagnetics literature). This is partly based on the excellent work on the authors on “conformal boundary optics” [28]. However, the manuscript gives only one equation about it [Eq. (8)] and not enough related information for self-consistency. We suggest to reduce some parts related to the GSTC-FDTD, replacing them with appropriate references (comment 1), and instead use the available space to better describe the GSTC-FDTD + Coordinate

Transformation procedure, which is the real core and contribution of the paper.

I have performed this review with the help of a student working in this area, and we both have difficulties in understand the gist of the method and a number of specific elements. The key idea of the paper is to use the conventional rectangular Yee grid (in contrast to what is erroneously stated somewhere, see below) everywhere, without stair-case approximation of the curved surface, and that the curvature of the metasurface is completely absorbed in the coordinate-transformed susceptibility functions. The “conformal” GSTCs are applied locally with curvature information included exclusively in them and not anywhere else. This fact and precise description of corresponding steps are not clear enough in the paper. Moreover, several points lack clarity and accuracy, some of which are mentioned below.

We are highly grateful to the reviewer for addressing this point. After reducing the derivations of the normal FDTD and GSTC-FDTD equations for planar devices, we are giving a detailed discussion on the methodology and the design of curved metasurface from the electromagnetic boundary conditions under different coordinate systems. The curvature of the metasurfaces and its FDTD implication are treated by introducing virtual nodes along two directions across the boundary. We have added these discussions in the main article and the supplementary materials.

DETAIL COMMENTS

1) There are many grammatical errors. To be carefully revised and corrected.

We apologize for our grammar problems, which may mislead the understanding of the manuscript. We have checked the entire manuscript carefully and have rewritten some parts of the paper trying to correct most of these issues.

2) Lines 77-78 read: "...complex boundary conditions with curved interfaces have never been implemented". This is incorrect. GSTCs have been applied to cylindrical and spherical metasurfaces in the following two respective papers:

- M. Safari, A. Abdolali, H. Kazemi, M. Albooyeh, M. Veysi, and

F. Capolino, "Cylindrical metasurfaces for exotic electromagnetic wave manipulations," in 2017 IEEE International Symposium on Antennas and Propagation USNC/URSI National Radio Science Meeting, July 2017, pp. 1499–1500.

- arXiv:1710.00040v1 [physics.app-ph] 7 Sep 2017

This must be correct and these reference must be added. Their existence does not reduce in any ways the merits of the proposed method, since those 2 papers work only on canonical (circular cylinder and spherical) curved metasurface, whereas the proposed method can handle any curvature.

Indeed, as the reviewer is pointing out, cylinder and spheres have inherently conformal boundary conditions in cylindrical and spherical coordinate systems,

respectively. We have cited these previous works and included a sentence in the main manuscript to avoid any misunderstanding. We have also included a paragraph at the end of the paper to discuss this in more details. It reads: *“It is worth noting that it is indeed possible to calculate curved problems considering locally flat interfaces. However the local condition for matching incident and outgoing waves will have to be adjusted locally, i.e. adjusting the incident angle with respect to the surface orientation and recalculating the susceptibilities point-by-point along the interface. In essence, this is exactly the purpose of using the conformal boundary condition. With this method, one can calculate the photonic response of the interface in the surface coordinate –intrinsically flat in the interface coordinate system- and transform both incoming and outgoing electromagnetic fields and susceptibilities from the local to the global coordinate system. This method allows us to simplify the problem at the conceptual level and it significantly reduces the problem of coding locally the boundary conditions.”*

3) The aforementioned “any curvature” is not completely right. For instance in example 2 (sinusoidal metasurface), the ratio between the modulation height and structure pitch must be small enough to avoid wave trapping and subsequent multiple reflections. A note on such restrictions should be included!

The reviewer highlights an interesting subwavelength regime that has never been considered in the original manuscript. Looking for wave trapping (resonators for examples) and subsequent multiple reflections on the curved metasurface stimulates appealing additional research directions. To avoid this regime, the oscillation periods

of all of our surface function have been chosen so that the oscillations are larger than the wavelength of incident light. A discussion on this interesting regime has been included in the manuscript and we thank the reviewers for pointing out this.

4) Detail: note, in relation to the synthesis (7), that diffraction-less refraction cannot be achieved in a monoanisotropic metasurface, as shown in arXiv:1705.09286v2 [physics.class-ph] 12 Jun 2017: a bianisotropic metasurface is required for that. This is a detail that does not affect the essence of the paper (on the contrary, the result show the spurious diffraction order, weak here due to the small angle), but this may be noted by the authors.

We thank you for your valuable suggestion. Indeed a bi-anisotropic metasurface is required to achieve diffraction-less refraction. We have mentioned this point in the revised version of the manuscript.

5) Eq. (8): $a = ab$ is impossible unless $a = 0$. Please, use different notation for the left-hand side χ and the χ involved in the right-hand side.

Yes, the reviewers caught a small but important typo in Eq8 (now Eq 4 in the revised manuscript), which has been corrected. Thank you for your careful proof reading.

6) Fig. 1: draw the local coordinate system for convenience. Also correct coordinate inconsistencies between (a) and (b).

We thank the reviewers for their suggestion and also for carefully checking the coordinate systems presented in Fig.1. The orientation of the local (tangent to the surface) and global coordinate systems have been included in the figure for a better illustration according to the FDTD decomposition in Equations. S1 to S4.

7) Lines 223-262: correct the order of the superscripts.

The order of the superscripts describing the susceptibility coordinate transformation were swapped. We thank the reviewers for highlighting this typo. We have carefully checked the rest of the manuscript and all examples to correct this issue.

8) Line 400 and other places in the paper refers to “modified Yee cell”. This is wrong, or at least misleading (see general comment 2 above). The Yee grid/cell is unchanged; only the “welding” update equations between REGULAR Yee cells are changed.

Indeed, this phrase might be misleading. In the planar metasurface description, the Yee cell is unchanged and the Equations (S5), (S6), (5), and (6) are implemented by inserting virtual nodes around to the normal Yee cell. We have corrected the misleading description. In the conformal description, we are still relying on Yee cell calculation but adding additional virtual nodes in the vicinity of curved metasurface to address its nonplanar geometry.

9. Examples should be described with a bit more clarity.

The reviewers are asking for further details on the examples. We have included the susceptibilities values of each examples in supplemental materials Fig S2 and S3.

We thank the Reviewer for this comment.

We have clarified some points and made further alterations as suggested by the Reviewer. We thank the reviewer for his valuable comments and his suggestions.

They have tremendously help us improving the readability and the clarity of our manuscript.

***Response to the comments of Reviewer 2.**

In this paper, the authors proposed a modified finite difference time-domain (FDTD) approach to calculate the electromagnetic fields across conformal metasurfaces of arbitrary shapes. The reviewer believes it is a good effort to investigate new computational methods to extend conventionally metamaterial simulation from Euclidean space to Riemann geometry. The proposed method may be utilized for the simulation and design of conformal metasurfaces. The reviewer recommends accepting the paper for publication in Nature Communications after the following problem has been fixed.

1) The authors should better elaborate the advantages of the modified FDTD methods versus the conventional FDTD method. Since its introduction by Yee, FDTD has already been widely used to obtain numerical solutions of Maxwell's equations for a broad range of problems. In the current content, it is not very clear what new improvements have been made in the modified FDTD.

We thank the reviewer for the positive comments on our manuscript and for appreciating the extension of conventional metamaterial simulation from Euclidean

space to Riemann geometry.

The conventional FDTD method can solve numerical Maxwell equations by substituting the electromagnetic parameters of the materials and the geometries of the whole structure. However, describing and calculating the entire photonic response of metasurfaces, i.e. computing all nanostructures, using conventional FDTD method and conventional design approach require huge computational time and is very costly in memory. The size of the FDTD mesh have to be considerably small to properly account for the geometry of nanoscale elements. Our modified FDTD method treats metasurface, a 2D counterpart of 3D metamaterials, as a zero thickness interface with the surface electric and magnetic susceptibility tensors. These quantities represent the photonic responses of an ensemble of nanostructures, which however cannot be immediately used and calculated in the conventional FDTD scheme. uses additional virtual nodes across the interface to address the discontinuity of the electromagnetic fields. We have rewritten the introduction part to elaborate the advantages of the presented modified FDTD method.

2) The authors claim that the proposed method can "simplify and accurately calculate the electromagnetic fields across conformal metasurfaces of arbitrary shapes". It is not clear to the reviewer how the proposed method can model complex structures using regular lattice grids and how to capture the discontinuity of material properties across the interface of different materials in the

metasurfaces. This is an fundamental issue in the design of multimaterial metasurfaces. It will be very helpful if the authors can better elaborate it with an example.

Using conformal boundary conditions, it is not anymore necessary to calculate the photonic response of a large number of nanostructures as it is generally the case for metasurfaces (see reply 1). We directly address the entire photonic response of the interfaces without having to model complex structures using regular lattice grids in the conventional FDTD method or other simulation software. Then, the response of the free form metasurface is fully addressed with the coordinate-transformed susceptibility functions. In all of our examples, metasurfaces are placed in the free space, meaning the simulation area is air-metasurface-air. To answer the reviewer comment on multimaterial case, we did a new simulation by considering the photonic response of a metasurface acting as a lens (Fig. 4(a)) deposited at the interface of air and silica (with refractive index 1.5). The simulation results showing the magnetic field intensity distribution are presented in the supplemental material figure S4, which demonstrates that our method is also applicable to the design of multimaterial metasurface cases.

3) Realistic FDTD simulations involve fine discretization of the spatial domain as well as the temporal domain. It is not uncommon to use a huge number of spatial grids and spatiotemporal grids. As a result, FDTD simulations require significant computational resources both in terms of memory and execution. It would be helpful if the authors can better elaborate the efficiency of the proposed method and whether it is scalable to large-scale problems.

We take Fig. 4(a) as an example. The simulation performed using our modified FDTD method takes 187MB memory and its execution takes 46 seconds with 1016*1016 meshes. Using conventional FDTD method and considering an array of sub-wavelength nanostructure representing the metasurfaces (typical size around 50nm large and 300nm tall), the simulation takes 1.7GB memory and 4 minutes execution time with 3016*3016 meshes to obtain similar results obtained by using our modified FDTD method. Both simulations are implemented in two-dimensional. The required computational resources would further increase for conventional FDTD

method for accurate calculation of metamaterials, which needs three-dimensional nanostructures calculations. Thus, it is the reason why we believe that our method would be more scalable to large-scale problems than the traditional FDTD methods.

We would like to thank the reviewer for the valuable comments that, we believe, helped us improving our manuscript.

***Response to the comments of Reviewer 3.**

The authors propose a new numerical method based on modified finite difference time domain approach that simplifies and accurately calculate the electromagnetic fields across conformal metasurfaces or arbitrary shapes. To validate this method, the authors propose 3 different examples such as absorbing metasurfaces, beam refractor, and curved lens, showing consistent results in agreement with fully theoretical predictions.

Although the paper is interesting, in my opinion, it does not introduce any novel phenomenon and does not provide any performance or technological advancement with respect to previous works based on standard FDTD simulations by using dense mesh, I believe it does not meet the impact and novelty requirements to be published in Nature communications journal.

It is well known that the conventional FDTD need to be refined when dealing with curved surfaces for accurate results. In fact, modeling dispersive materials with curved surfaces still remains a challenging topic, not only because the algorithm dealing with it is complex, but also because it can become numerically unstable. One approach to addressing this problem is based on the use of effective permittivities (EPs). This method is well known in the literature (see reference below).

M. G. Silveirinha, A. Alu, and N. Engheta, "Parallel-plate metamaterials for cloaking structures," Phys. Rev. E, 75, 036603 (2007).

G. W. Milton and N. P. Nicorovici, "on the cloaking effects associated with

anomalous localized resonance,” *Proc. R. Soc. A*, **462**, 3027--3059 (2006).

D. Schurig, J. B. Pendry, and D. R. Smith, “Calculation of material properties and ray tracing in transformation media,” *Opt. Express*, **14**, 9794--9804 (2006).

On the other hand, this kind of structure is well understood, and a numerical and experimental paper has been published recently in *Nature Communications* without using this method.

“Decoupling optical function and geometrical form using conformal flexible dielectric metasurfaces” *Nature Communications* **7**, 11618 (2016)

In my opinion, the term “a new and powerful design tool urgently needs it” is exaggerated.

In order to improve and give more impact to the paper for other journals like the journal of computational physics, physical review E.

We thank the reviewer for dedicating the time and the effort of reading our manuscript. We also thank the reviewer for finding our manuscript interesting.

The main novelty of this work as compared to the conventional FDTD methods is that we propose a general design method for functional metasurface (a 2D counterpart of 3D metamaterials) devices with arbitrary geometry and much-reduced computational load.

We have rewritten a considerable part of the manuscript, starting from the discussion on the analytical models using GSTC and conformal boundary optics. We have modified this introduction to better emphasize the unique 2D description of metasurfaces with respect to 3D metamaterials. As the reviewer mentioned, the

effective permittivity are usually employed to model three-dimensional metamaterials. Note that in our GSTC-based approach model for calculation of conformal metasurfaces, the optical response of metasurface is characterized by electric and magnetic surface susceptibilities. The latter are not bulk quantities and cannot be calculated using the traditional effective medium theory. Compared to conventional FDTD algorithms, our approach modifies the FDTD algorithm only in the vicinity of the curved metasurfaces by creating virtual notes to address the discontinuity of the electromagnetic fields, thus allowing us to calculate easily the transmission and reflection of arbitrary wavefronts. To highlight better the novelty of our algorithm, we have considerably modified the introduction and have discussed the connection of this numerical approach with the formal description of conformal boundary conditions.

Reviewer 3 is asking an important question regarding the performance of this method compared to usual metasurface simulation approach. This question somehow overlap with reviewer 2's comment. To compare the performance, we considered the example in Fig. 4(a). The simulation performed using our modified FDTD method takes 187MB memory and its execution takes 46 seconds with 1016×1016 meshes. Using conventional FDTD method and considering an array of sub-wavelength nanostructure to represent a metasurface (typical size around 50nm large and 300nm tall), in order to get similar results, we have to use around an order of magnitude more memory and 4 minutes execution time with 3016×3016 meshes.

I suggest the authors address the following questions.

1) In this paper, equation 7 presents the relation between surface electric and magnetic susceptibility tensor and reflection and transmission coefficients. Although it is a common equation in GSTCs theory, I still suggest the authors provide a reference. Besides, this equation is only for normal incidence. I also suggest the author mention it in the paper or expand the method to oblique incidence

The relation of the surface electric and magnetic susceptibility tensor and reflection and transmission coefficients have been discussed in Refs. 26, 27, which are added in the revised manuscript. We have properly cited and highlighted these pioneering works in our manuscript. Reflection and transmission coefficients as calculated and presented in Eq 1 in the revised manuscript actually use arbitrary incident angles. Because the electric fields are expressed in this equation, the susceptibilities are automatically satisfied for various incident angles. Obviously, the susceptibilities change as a function of the incident angle. Therefore, all complex susceptibility values in equation 7 (Eq. 1 in the revised version) are suitable for all incident angles and in fact any incident wavefronts. This relation is also used in other reference; for example Refs. 34-38.

2) In this paper, the authors demonstrated three examples, absorber, deflector, and concentrator with flat case and curved boundary condition. Also, the curved boundary condition can be also solved by locally flat boundary condition with

small mesh. In order to see the difference between the old (flat) and the new (curved) method, I suggest the author compare the results of them.

In principle, the reviewer is correct that the curved boundary condition may be also solved by the locally flat boundary condition with small mesh but the computation would be extremely complicated for the latter case. The key advantage of our proposed method is the use of conformal boundary conditions. We apologize that we didn't make this point clear in our original manuscript. We have modified considerably the introduction and have included the following sentences to summarize our approach (Note that this point was also raised by reviewer 1, the second comment). *"It is worth noting that it is indeed possible to calculate curved problems considering locally flat interfaces. However the local condition for matching incident and outgoing waves will have to be adjusted locally, i.e. adjusting the incident angle with respect to the surface orientation and recalculating the susceptibilities point-by-point along the interface. In essence, this is exactly the purpose of using the conformal boundary condition. With this method, one can calculate the photonic response of the interface in the surface coordinate – intrinsically flat in the interface coordinate system- and transform both incoming and outgoing electromagnetic fields and susceptibilities from the local to the global coordinate system. This method allows us to simplify the problem at the conceptual level and it significantly reduces the problem of coding locally the boundary conditions."* The reviewer is actually highlighting the key contribution of this work, in which one can address the problem of local versus global computation method on

conformal interfaces. We thank the reviewer for making us aware that we needed to explain better the purpose of using conformal boundary conditions.

3) In figure 3(b), the variation of the wave front is much larger than the flat case in figure 3(a). I suggest the authors decrease the mesh size. It is also the same case in figure 4(b).

This problem is not related to the mesh size but rather due to the assumption that we have made on the susceptibilities. For convenience, we consider the simplest expressions for the representation of our metasurfaces, i.e. mono-anisotropic susceptibilities. As it has been correctly pointed out by the reviewer 1, pure plane wave are perfectly transmitted/reflected only when the surface susceptibilities are bi-anisotropic. The use of mono-anisotropic tensors is correct but creates some diffraction that induces small variation of the wavefront with respect to pure plane wave. This problem cannot be simply solved by decreasing the mesh size as suggested by reviewer 3. Bi-anisotropic response can be considered in our model and we have added this discussion in the manuscript.

We thank the reviewer for his valuable comments and his suggestions. They have tremendously help us improving the readability and the clarity of our manuscript. After resolving these important issues, we hope that the reviewer appreciate better the research discussed and reported in our manuscript.

Reviewers' comments:

Reviewer #1 (Remarks to the Author):

More comments:

- 1) Abstract: GSTCs are NOT only to describe flat optical devices (e.g. reported cylindrical and spherical ones).
- 2) lines 78-79: "Arbitrary wavefront control requires balanced control of loss and gain, and these quantities are generally well-captured using GSTC models." This is incorrect; there are other - and better ways! - ways, including bianisotropy engineering.
- 3) Lines 106 and 107: Proper reference should be provided to previously reported concept of "virtual nodes."
- 4) Eq. (5) is not clear: it is an FDTD, i.e. TIME-DOMAIN equation, but includes... the angular frequency ω !?! How could this equation be handled in the case of a Lorentz dispersion (see literature solution)?
- 5) Line 197 and others: give sizes not in absolute terms (little meaning) but relatively to the wavelength.
- 6) Lines 281-286: As showed in the literature, this (diluted slab simulation) should lead to an exact result.
- 7) Reference 38 is not available on IEEE and should be updated.

Reviewer #2 (Remarks to the Author):

I recommend accepting the revised paper for publication in Nature Communications considering the fact that the authors have addressed most of the comments from the reviewer.

Reviewer #3 (Remarks to the Author):

I would first like to thank the authors for dedicating time and efforts to respond to all questions. The claims are now well supported by numerical evidence and good comparison. The manuscript is now well-written and it is easy to follow.

However, although the paper is interesting, in my opinion, it does not introduce any novel concept to justify publication in Nature Communications. Compared to the previous work (reference 32 in this manuscript) which numerically and experimentally demonstrated curvature metasurfaces, the only novelty the authors claim in this paper is from a flat surface to a curved surface. The standard metasurfaces (flat metasurfaces) papers based on GSTCs are usually published in journals like IEEE (references 27, 28 in the manuscript). Besides, the similar curved metasurface problem has also been investigated and published in Phys. Rev. A by the same authors (reference 29 in the manuscript).

I remain deeply convinced that the paper deserves to be published in computational physics journals or Physical Review E or Physical Review A as done for their previous work (reference 29 in the manuscript) which is more relevant than the current paper.

I should however note that, reading the reviews from other referees, I will accept the editorial decision on the current work.